# Theoretical Design of a Bionic Spatial 3D-Arrayed Multifocal Metalens

**DOI:** 10.3390/biomimetics7040200

**Published:** 2022-11-16

**Authors:** Guihui Duan, Ce Zhang, Dongsheng Yang, Zhaolong Wang

**Affiliations:** 1Interdisciplinary Research Center of Low-Carbon Technology and Equipment, College of Mechanical and Vehicle Engineering, Hunan University, Changsha 410082, China; 2Qian Xuesen Laboratory of Space Technology, China Academy of Space Technology (CAST), Beijing 100094, China; 3Beijing Spacecrafts, Beijing 100094, China; 4Jiangsu Key Laboratory of Micro and Nano Heat Fluid Flow Technology and Energy Application, Suzhou University of Science and Technology, Suzhou 100094, China

**Keywords:** bionics, 3D-arrayed multifocal metalens, nanofin, geometric phase, polarization-dependent

## Abstract

With the development of micro/nano-optics, metasurfaces are gaining increasing attention working as novel electromagnetic wave control devices. Among which, metalenses have been developed and applied as a typical application of metasurfaces owing to their unique optical properties. However, most of those previous metalenses can only produce one focal point, which severely limits their applications. Inspired by the fly compound eye, we propose a special kind of spatial multifocal metalens. Our metalenses can reverse the polarization state of the incident circularly polarized light, which is then focused. In addition, a horizontally aligned multifocal metalens can be achieved by designing reasonable phase and region distributions, which is similar to a vertically aligned one. Most significantly, a spatially 3D-arrayed multifocal metalens with low crosstalk is well achieved by combining these two distribution methods. The proposed bionic 3D-arrayed multifocal metalens with amazing focusing effect promises applications in imaging, nanoparticle manipulation, optical communication, and other fields.

## 1. Introduction

Metamaterials are structured surfaces that do not exist in nature, and their physical properties are mainly determined by subwavelength structures [1]. Therefore, metamaterials exhibit numerous novel physical properties, such as negative refractive index [2], inverse Doppler effect [3], inverse Cherenkov radiation [4], perfect solar absorption [5,6,7,8], and so on. However, the limitations of those existing fabrication techniques make it difficult to fabricate three-dimensional (3D) metamaterials [9,10]. Therefore, the two-dimensional (2D) metamaterials, metasurfaces, have been rapidly developed and applied owing to their relatively easy fabrication methods and technologies [11,12,13,14,15,16]. Moreover, metasurfaces can control the amplitude [17,18,19,20], phase [21,22,23,24], and polarization [25,26,27,28] of electromagnetic waves, leading to a fact that various applications based on their unique properties have been developed, such as metalens focusing [29,30,31], holographic imaging [32], vortex light modulation [33], and abnormal deflectors [34,35].

As the most typical application of metasurfaces, metalenses utilizes the micro-nano structures to control light, who regulates light in multiple dimensions, which is different with traditional refractive lenses [36]. Thus, a metalens based on diffractive optics can be as thin as hundreds of nanometers, allowing the fabrication of ultrathin flat optical devices with greatly reduced size and mass [37]. Since the first metalens proposed by Federico Capasso in 2016 [37], metalenses have now been well developed with high numerical aperture (NA) [38], high resolution [39], high efficiency [40], achromaticity [29,31,41], new functions [42], etc. The phase manipulation methods of metasurfaces can be divided into propagation phase [43], resonance phase [44], geometric phase [45], etc. However, there is only one focal point for most of the previous metalenses, and the applications of those single focal point metalenses are limited in many scenarios. The multifocal metalenses focus incident light at different focal points in the horizontal and vertical directions with potential applications in imaging systems [46], particle manipulation [47], and optical communication [48], thus they should be developed for the multifunctional integration.

Inspired by the fly’s compound eye with numerous focal points [49], we theoretically propose a spatially multifocal metalens. A phase manipulation method with nested square rings to design a polarization-dependent multifocal metalens based on the geometric phase principle is employed. With such an arrangement of the nanostructures on the top, multiple focal points of the metalens can be obtained in both of the vertical and horizontal directions. Simulation results demonstrate that our designed metalenses can realize the integration of multiple focal points and effectively reduce the crosstalk between the subfocal points. Moreover, the intensity ratio of each subfocal points can be adjusted by further changing the arrangement mode. Most significantly, the proposed bionic 3D-arrayed multifocal metalens combining of vertical and horizontal focal point arrangements exhibits amazing focusing effect, promises applications in the fields of high-resolution 3D imaging, nanoparticle manipulation, and so on.

## 2. Structure Design and Analysis

The fly compound eye is schematically shown in Figure 1a; it consists of an array of microeyes distributed in the *xy* plane, and every microeye is equivalent to a microlens with a unique focal point distributed in a three dimensions space [49]. Inspired by such a unique structure with numerous focal points, we propose a multifocal metalens as shown in Figure 1b. Each sublens arranged in the *xy* plane is equivalent to a microeye in the compound eye as shown in Figure 1a, and the size of our sublenses is 50 μm × 50 μm. In addition, three types of metalens are proposed step-by-step: first, a metalens with focal points distributed in the *xy* plane is designed. Then, focal points of a metalens distributed in *z* direction will be achieved. At last, a multifocal metalens with focal points distributed in three dimensions similar to the fly compound eye is proposed by combining these two previous kinds of distributions of focal points based on the sublenses arranged in *xy* plane.

The proposed bionic metalenses consist of nanofins as the unit cell for phase modulations of different polarized light. The equivalent refractive indices along the two intersecting axes of these non-centrosymmetrical nanofins are different for polarization state multiplexing. Considering the refractive index and energy loss, we choose titanium dioxide (TiO_2_) instead of metallic materials for the nanofins on the top of the proposed bionic metalenses, and the substrate is made from silicon dioxide (SiO_2_). The refractive indices of TiO_2_ (*n*_1_) and SiO_2_ (*n*_2_) are as high as 2.668 and 1.547 at 532 nm, respectively. In contrast, their extinction coefficients are close to zero, which ensures the ultra-high transmittance of the proposed metalenses. The unit cells of TiO_2_ nanofins are long square cylinders, which strongly depend on their geometric parameters of length (*L*), width (*W*), and height (*H*). A 3D model of a single nanofin is shown in Figure 2a, wherein the light blue structure represents SiO_2_ substrate and the gray structure represents TiO_2_ nanofins. To completely reverse the polarization state of the incident light, the unit cells are designed as a half-wave plate.

Furthermore, we sweep the *L* and *W* using finite-difference time-domain (FDTD) method based on Maxwell equations [6,50] to make the nanofins meet the phase change requirements of 0–2*π* by setting *H* = 600 nm and period *S* = 400 nm. The transmission phases of the incident light vary from −*π* to *π*. The transmission phases of the nanofins for two orthogonally incident linearly polarized (LP) lights are simulated by changing *L* and *W*, and the results are illustrated in Figure 2b and 2c, respectively. The difference in the transmission phase of the incident LP light along the *x* axis and the *y* axis is shown in Figure 2d. It can be observed that such a phase difference is also diagonally symmetric due to the axial symmetry of nanofins. The phase difference produced by two orthogonal LP lights is *π* when *W* = 110 nm and *L* = 230 nm (the position of the red dot in Figure 2d), which satisfies the condition of a half-wave plate. Therefore, these geometric parameters are set as default for our bionic metalenses.

We use nested square ring regions to generate different phase distributions to fully utilize the space of the proposed metalenses. As shown in Figure 2e, the two different colored areas with a respectively focal length of 50 μm (violet) and 100 μm (blue) are used to generate the predicted phase distributions. In addition, more focal points of a bionic metalens can be achieved by nesting more levels of square ring. We also numerically study the characteristics of the geometric phase of our bionic metalenses (Figure 2f). The phase angle is twice the rotation angle of the nanofins based on the geometric phase principle, which is in good agreement with our simulation results. Furthermore, we theoretically calculate the radial direction phase distribution of metalenses with different focal lengths, and the results are shown in Figure 2g. Their NAs are 0.707 (*f* = 25 μm), 0.447 (*f* = 50 μm), 0.316 (*f* = 75 μm), and 0.243 (*f* = 100 μm), respectively. It can be concluded that the shorter the focal length of the same size metalens, the more orders of the phase that can be modulated.

The phase distribution of a proposed metalens should be similar to a spherical lens in order to obtain the same functions. The specific calculation equation [51] for the transmission phase (*φ*) of a metalens is as follows:(1)$$\varphi =\frac{2\pi }{\lambda }(\sqrt{\left(x^{2}+y^{2}+f^{2}\right)}-f)$$
where *λ* is the wavelength of the incident light (*λ* = 532 nm), (*x*, *y*) is the central coordinate of each nanofin, the geometric center of the metalens is at (0, 0), and *f* is the focal length of the metalens. Therefore, we can calculate the phase for each nanofin only by the wavelength of the incident light, coordinate position, and the focal length of the nanofins. According to the relationship between the phase value (*φ*) and the rotation angle (*θ*) of the nanofins:(2)φ=±2θ
where “±” represents the polarization states of right-hand circularly polarized (RCP) light and left-handed circularly polarized (LCP) light, respectively. So, the rotation angle of each nanofin can be calculated for RCP in the metalens as follows:(3)$$\theta =\frac{\pi }{\lambda }(\sqrt{\left(x^{2}+y^{2}+f^{2}\right)}-f)$$

Furthermore, the rotation angle of each nanofin for LCP in the metalens can be calculated as follows:(4)$$\theta =\frac{\pi }{\lambda }(f-\sqrt{\left(x^{2}+y^{2}+f^{2}\right)})$$

Therefore, once the polarization state of the incident circularly polarized light is determined, only the coordinate position of each nanofin, the wavelength of the incident light, and the focal length of the metalens are needed to calculate the target phase and rotation angle of each nanofin.

## 3. Results and Discussions

We firstly design bionic metalenses with one, two, three, and four focal points in the vertical direction, respectively. The maximum focal lengths of those four focal points of the proposed bionic metalenses are respectively 25 μm, 50 μm, 75 μm, and 100 μm, and their phase distributions are all the nested square ring shown in Figure 2e. The plane where the upper surface of the substrate located is taken as the *xy* plane (*z* = 0) of the spatial rectangular coordinate system. Therefore, the focal point coordinates are (0, 0, 25), (0, 0, 50), (0, 0, 75), and (0, 0, 100), respectively. The light intensity distributions of the multifocal metalens along the optical axis and that on the *xz* plane (*y* = 0) are shown in Figure 3a–d. For the single focal point metalens shown in Figure 3a, its focal length (*f*) is 25 μm with a metalens size of 50 μm, and the ambient medium is air with a refractive index (*n_air_*) of 1. Also, the NA is 0.707, and its full width at half maximum (FWHM) at the focal point is 1.4 μm. When the number of focal points increases, the energy of the incident light will be dispersed, and the larger the focal length, the smaller the NA but the larger the focal spot as well as the focal depth. The results also indicate that the crosstalk between subfocal points along the *z* axis is very low, which ensures the independence of these focal points.

For the same polarization state of incident circularly polarized light at the same wavelength, we only need to select the appropriate coordinates and substitute the designed focal length to calculate the target phase distribution. Thus, we can design multiple focal points on the optical axis, which are only limited by the size of the metalens and the discrete distribution of the unit cells. We design a multifocal metalens with four focal points in the same horizontal plane as shown in Figure 2f.

In the horizontal direction, all the focal lengths of the multifocal metalens are 75 μm, leading to a fact that their coordinates are (12.5, 12.5, 75), (−12.5, 12.5, 75), (−12.5, −12.5, 75), and (12.5, −12.5, 75), respectively. However, the optical axis of each sublens will have an inclined angle along the *z* axis since the center coordinate of the lens is at (0, 0, 0). Considering the symmetry of the multifocal metalens, we only need to analyze the light distribution of the focal point at (12.5, 12.5, 75) because those of the four are the same. The light intensity distributions of a multifocal metalens along the *z* axis and that on the *xy* plane (*z* = 75) are shown in Figure 3g,h, respectively. As shown in Figure 3, the proposed multifocal metalens exhibits good focusing effect with reasonable phase and region distributions. However, the overall focusing effect of the metalens begins to decrease because of the increase of the background noise when the number of focal points increases. Also, more energy of the incident light is dispersed due to the crosstalk between multiple focal points. In addition, the crosstalk between two adjacent focal points becomes more serious when the distance between two adjacent focal points is shortened. It can be clearly seen that the intensity of the focal points is higher with shorter focal depth when the focal point is closer to the metalens. For the horizontal multifocal metalens with the nested square ring structures, the optical axis of the sublens will produce an included angle in the vertical direction due to the deviation of the focal points.

For incident light with different polarization states, the phase distribution needs to be additionally modified. In the present study, the polarization states of the incident light are set to RCP light, LCP light, and LP light, respectively. The focal lengths are 50 μm (RCP light) and 100 μm (LCP light), with their corresponding coordinates of (0, 0, 50) and (0, 0, 100), respectively. For LP light, the polarization state can be decomposed into orthogonal RCP light and LCP light. Therefore, a focal length of 50 μm for the designed metalens is achieved with RCP light (Figure 4a), while a focal length of 100 μm is realized for LCP light (Figure 4b), and focal lengths of a metalens of 50 μm and 100 μm are realized simultaneously when LP light is incident (Figure 4c). The light intensity distributions of these multifocal metalenses along the optical axis and those in the *xz* plane are shown in Figure 4. It can be observed that different focal lengths and numbers of focal points are realized when the light incident on the designed multifocal metalens has different polarization states. Moreover, the axial light intensity distribution is almost equal to the superposition of LCP and RCP lights for LP light because the LP light can be split into two orthogonal circularly polarized lights. Furthermore, such a multifocal metalens can be used in scenarios in which multiple polarization states should be simultaneously controlled.

In addition to the multiple focal points arranged along the vertical or horizontal direction, we also discuss a metalens with different focal lengths on different planes in space for potential applications of high-dimensional and multiphoton quantum source [52], as well as near-eye integral imaging [43]. As the projection point of each focal point on the *z* = 0 plane is not at the origin of the Cartesian coordinates, the phase calculation formula for each subfocal point is also different. Based on Equation (1), the rotation angle of each nanofin can be calculated by substituting the center coordinates and spatial distribution of each sublens designed. The specific calculation equation is as follows:(5)$$\theta =\frac{\pi }{\lambda }(\sqrt{\left({\left(x-x\prime \right)}^{2}+{\left(y-y\prime \right)}^{2}+f\prime^{2}\right)}-f\prime )$$
where (*x*′, *y*′) is the center coordinates of each sublens and f′ is the focal length of each sublens.

As shown in Figure 5a,b, four small squares represented by four different colors are divided from a large square region with a length of 50 μm. Every small square will act as a sublens with focal lengths of 25 μm, 50 μm, 75 μm, and 100 μm, respectively. Therefore, the coordinates of these four focal points are (12.5, 12.5, 25), (−12.5, 12.5, 50), (−12.5, −12.5, 75), and (12.5, −12.5, 100) in the Cartesian coordinates.

As shown in Figure 5d–I, there is a good focusing effect when the focal length of a sublens is 25 μm, and the focusing effect of the sublens becomes worse and worse with the increase of its focal length (Figure 5d). The reason is that it is difficult to meet the requirements for phase ranges in 0–2π when the sublens is too small. For example, the size of each sublens is only 25 μm × 25 μm in the present metalens. The focal spots are seriously scattered when the focal lengths of the sublens are 75 μm and 100 μm, and the normalized light intensity is even lower than 20% compared with that of a sublens with a focal length of 25 μm.

Therefore, to enable the bionic multifocal metalens to focus efficiently at multiple points in space, we use the region division method as shown in Figure 2e and change the number of levels to 4, and the focusing effects of such a metalens are shown in Figure 6a. Every nested square region satisfies the targeted phase distributions of one focal point (Figure 6b), and the normalized light intensity distributions along the optical axes and those on the *xz* plane are shown in Figure 6c,d, respectively. In the Cartesian coordinates, the coordinates of these focal point are respectively (12.5, 12.5, 25), (−12.5, 12.5, 50), (−12.5, −12.5, 75), and (12.5, −12.5, 100), and the center coordinate of each sub-lens can be regarded as (0, 0, 0), so their optical axes are inclined as shown in Figure 6d. By comparing Figure 5 and Figure 6, it can be observed that the multifocal metalens shown in Figure 6 exhibits a better focusing effect. The light intensity of each focal point is more reasonable, and the crosstalk between the adjacent spots is much smaller. Furthermore, it can be demonstrated that different regional distributions still slightly affect the focusing properties of the multifocal metalens even if the size and focal points of each metalens are the same, but the focusing effect of multifocal metalens can be optimized through reasonable design. By comparing the intensity distribution diagrams in Figure 5 and Figure 6, it is clearly observed that the nested square rings presented a better multi-focus focusing effect than those others.

A bionic 3D-arrayed multifocal metalens is further developed by combining the abovementioned horizontal and vertical multifocal metalenses designed with 1D and 2D arrangements. A schematic diagram of such a metalens is shown in Figure 7a. The area of this metalens is 100 μm × 100 μm, and there are a total of 16 focal points in four planes (*z* = 25 μm, 50 μm, 75 μm, and 100 μm, respectively). It can also be observed from Figure 7b–e that the light intensities at the four focal points decrease along the positive *z* direction, and the crosstalk between two adjacent focal points also increases with the increase of the focal length. The results are similar to the focusing properties of our vertical multifocal metalenses shown in Figure 3. Most significantly, so many focal points in a limited space can greatly improve the efficiency of space utilization, which is critical for high-cost and highly difficult micro-nano fabrication processes. Such a multifocal metalens can be fabricated using electron beam lithography (EBL), atomic layer deposition (ALD) [14,53,54], and ion beam polishing (IBP) [55].

## 4. Conclusions

Inspired by the fly’s compound eye, we propose a multifocal metalens based on the geometric phase principle in this paper. Nanofins are chosen as the basic unit cell for the metalens. The length and width of nanofins are firstly swept and optimized to create a half-wave plate. Then, every nanofin is rotated along the *z* axis to form the wavefront similar to a spherical lens according to the targeted phase distribution. In addition, we also investigate the focusing effects of our metalens incident by RCP light, LCP light, and LP light at 532 nm. For incident light with different polarization states, multiple focal points along the horizontal and vertical directions can be achieved with good focusing effect. However, the overall focusing effect becomes worse with the increasing number of focal points for a multifocal metalens. Therefore, the appropriate region and phase distributions must be carefully selected to avoid affecting the focusing effects for a multifocal metalens with strong coupling effects of adjacent focal points. Most significantly, we propose a new strategy for designing 3D-arrayed multifocal metalenses with tens of focal points, exhibiting an amazing focusing performance similar to the fly’s compound eye, which promises applications of optical capture, particle acceleration, 3D imaging, and so on.

## Figures and Tables

**Figure 1 biomimetics-07-00200-f001:**
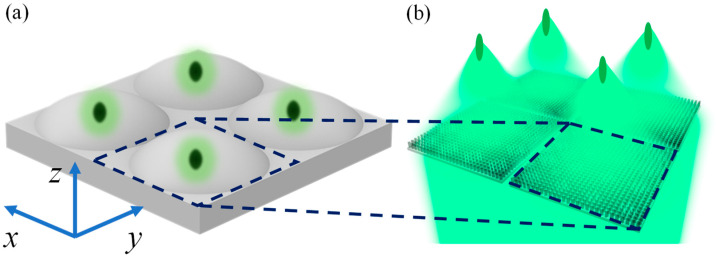
(**a**) Schematic diagram of a fly compound eye. (**b**) Multifocal metalenses array.

**Figure 2 biomimetics-07-00200-f002:**
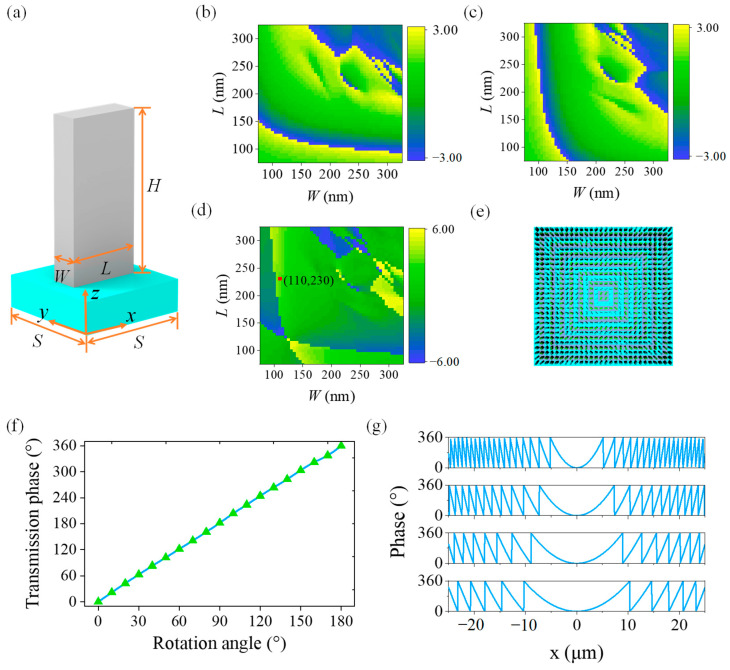
A multifocal metalens. (**a**) The 3D model of a unit cell, where *W*, *L*, *H* and *S* represent width, length, height and period size of a unit cell, respectively. Transmission phase of the polarization direction of the incident light along the (**b**) *x* axis and (**c**) *y* axis. (**d**) Phase difference of the polarization direction of the incident light along the *x* and *y* axis. (**e**) Nested square rings (different colors) of a metalens used to generate different phase distributions for different focal lengths. (**f**) Relationship between rotation angles of nanofins and the transmission phase. (**g**) Phase distributions of the bionic metalenses with different focal lengths (from top to bottom: *f* = 25 μm, 50 μm, 75 μm, and 100 μm, respectively).

**Figure 3 biomimetics-07-00200-f003:**
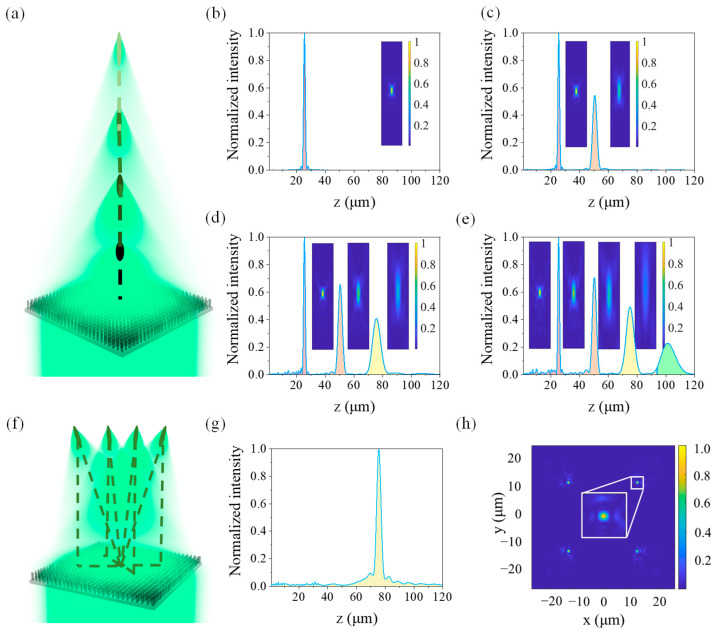
Multifocal metalenses with different focal points. (**a**) Four focal points are arranged in the vertical direction of a metalens. (**b**) A single focal point metalens (*f* = 25 μm). (**c**) A multifocal metalens with two focal points (*f* = 25 μm and 50 μm). (**d**) A multifocal metalens with three focal points (*f* = 25 μm, 50 μm, and 75 μm). (**e**) A multifocal metalens with four focal points (*f* = 25 μm, 50 μm, 75 μm, and 100 μm). (**f**) Four focal points are arranged in the horizontal direction (all their focal points are in the plane of *z* = 75 μm) of a metalens. (**g**) Intensity curve of a horizontal multifocal metalens along the *z* axis. (**h**) Normalized light intensity distributions on the *z* = 75 μm plane.

**Figure 4 biomimetics-07-00200-f004:**
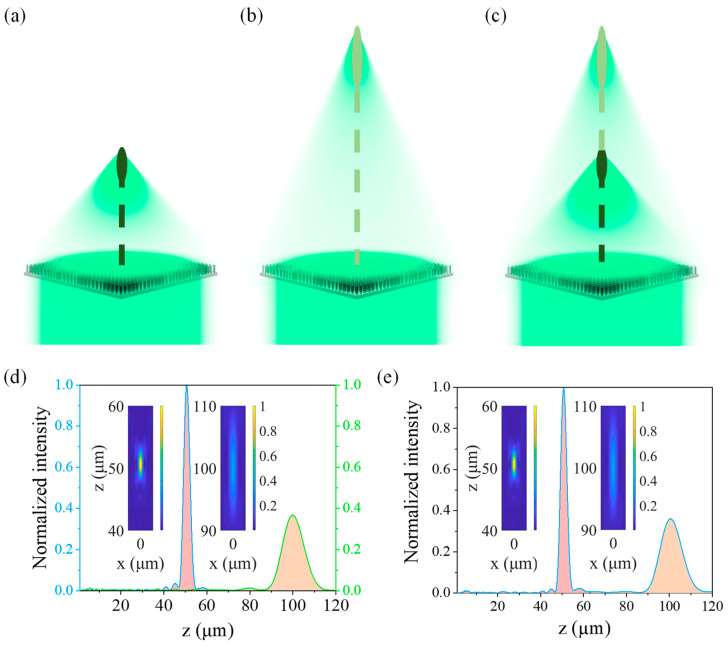
Multifocal metalenses for different polarized light. (**a**) RCP light. (**b**) LCP light. (**c**) LP light. Intensity distributions of the focal field for a multifocal metalens along the *z* axis and the vertical plane for incident of (**c**) RCP light (blue curve), (**d**) LCP light (green curve), and (**e**) LP light.

**Figure 5 biomimetics-07-00200-f005:**
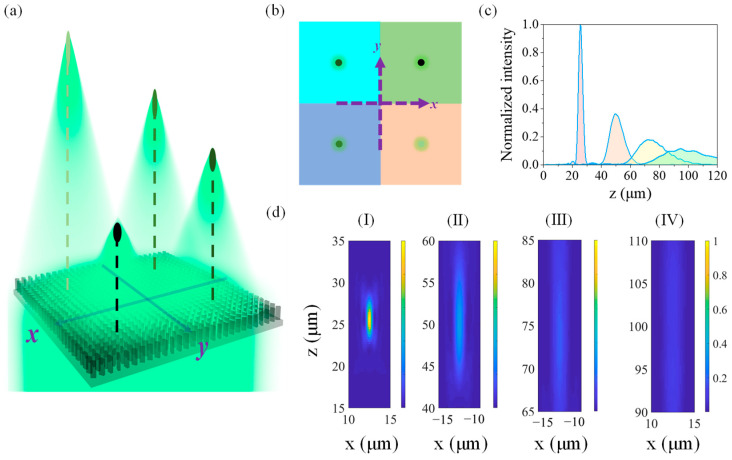
A metalens with different focal lengths on different planes in space. (**a**) A metalens with four focal points in space. (**b**) Regional distribution of the four sublenses and the projection of the focal points on the horizontal plane. (**c**) Light intensity distributions of each sublens of the multifocal metalens along the respective optical axes. (**d**) Intensity distributions of the focal field for a multifocal metalens in the *x*–*z* plane (*y* = 12.5 μm). (I) *x* ∈ [10, 15] and *z* ∈ [15, 35]. (II) *x* ∈ [−15, −10] and *z* ∈ [40, 60]. Intensity distributions in the *xz* plane (*y* = −12.5 μm). (III) *x* ∈ [−15, −10] and *z* ∈ [65, 85]. (IV) *x* ∈ [10, 15] and *z* ∈ [90, 110].

**Figure 6 biomimetics-07-00200-f006:**
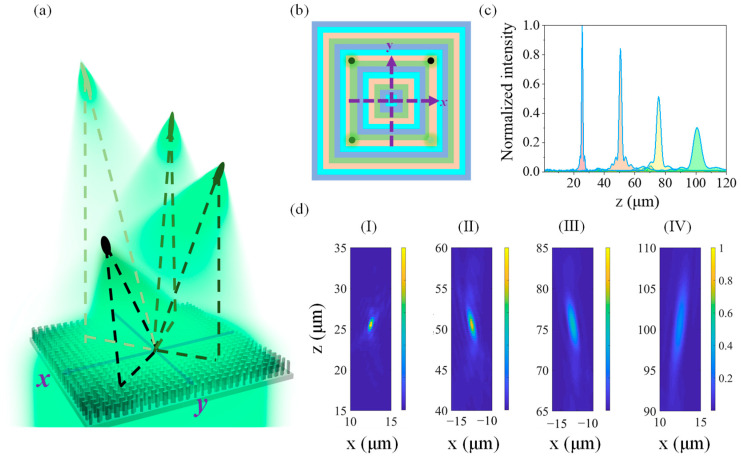
A metalens with multiple focal points in space. (**a**) Schematic of a multifocal metalens and its focal points. (**b**) Regional distributions of the four sublenses and the projection of the focal points on the horizontal plane. (**c**) Intensity distributions of the focal fields for a multifocal metalens along the respective optical axes. (**d**) Intensity distributions of the focal fields for a multifocal metalens in the *xz* plane (*y* = 12.5 μm). (I) *x* ∈ [10, 15] and *z* ∈ [15, 35]. (II) *x* ∈ [−15, −10] and *z* ∈ [40, 60]. Intensity distributions in the *xz* plane (*y* = −12.5 μm). (III) *x* ∈ [−15, −10] and *z* ∈ [65, 85]. (IV) *x* ∈ [10, 15] and *z* ∈ [90, 110].

**Figure 7 biomimetics-07-00200-f007:**
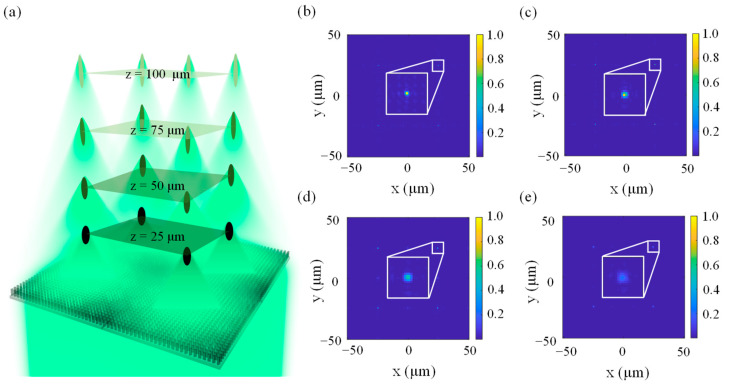
A 3D-arrayed multifocal metalens. (**a**) Schematic diagram of a spatial-array multifocal metalens. Distributions of light intensity in the *z* = (**b**) 25 μm, (**c**) 50 μm, (**d**) 75 μm, and (**e**) 100 μm planes, respectively.

## Data Availability

Not applicable.

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
