# Peer review of "Theoretical Design of a Bionic Spatial 3D-Arrayed Multifocal Metalens"

_biomimetics, 2022, doi:10.3390/biomimetics7040200_

Round 1
Reviewer 1 Report
From my point of view, the paper “Bionic spatial 3D-arrayed multifocal metalens” written by Guihui Duan, Ce Zhang, Dongsheng Yang and Zhaolong Wang is very interesting but difficult to follow, especially due to the passing from one type of multi-focal metalens to another one. I think that will be better a clear separation between them in such a way that the readers can understand better the results. Also, there are various dimensions for the sub-lenses and meta-lenses, therefore I suggest to the authors to make a clear distinction between the metalens dimensions and the dimensions of the constituent sub-metalenses.
- at the line 65 I think is a minor mistake - “microsye”
- at the lines 268-269 - the phrase “The results are consistent with the focusing properties of our previously designed vertical multifocal metalenses” is a little bit ambiguous – one cannot understand if it’s about the metalenses designed before in the same paper or if the authors are speaking about previous results published in another scientific paper.
Overall, I think the paper has potential and seems that the authors have paid attention to all the details. I suggest that can be accepted after minor arrangements.
Reviewer 2 Report
Review of manuscript entitled “Bionic spatial 3D-arrayed multifocal metalens.” By Guihui Duan et al.
The manuscript describes a theoretical work describing the behavior of a four element submetalenses. The position of the four focal points is studied under different situations. The compound lens is structured with minor components called nanofins. Period and height of nanofins is 600 nm and 400 nm respectively. Each of the four sublenses is 50 microns X 50 microns. Light could be right, left or linearly polarized.
Several comments are presented next.
1 The journal Biomimetics is more focused in biology. The only relation of the manuscript with biology is that it mimics the position of four lenses as the compound lens of a fly. I think the manuscript would be best positioned in an Optics journal.
2 The title does not mention that this study is from the point of view of theory. Thus I suggest to change the title by the following one:
Theoretical study of a bionic spatial 3D-arrayed multifocal metalens.
3 In line 52, 65 and 66 it is mentioned that the fly’s compound eye has numerous focal lengths. Give a reference where you get this information.
4 In line 121 an equation is shown, give a reference where it is mentioned.
5 In line 216 another equation is mentioned, give a reference for the equation.
6 In case somebody would like to fabricate this type of lens you could give some advices to them.
